# The Non-Cancer Specific Elevation of the Serum Squamous Cell Carcinoma Antigen during the Post-Radiotherapy Follow-Up of Cervical Cancer Patients

**DOI:** 10.3390/diagnostics11091585

**Published:** 2021-08-31

**Authors:** Tae Oike, Takahiro Oike, Ken Ando, Akira Iwase, Tatsuya Ohno

**Affiliations:** 1Department of Obstetrics and Gynecology, Gunma University Graduate School of Medicine, 3-39-22, Showa-machi, Maebashi 371-8511, Gunma, Japan; taeoike@gunma-u.ac.jp (T.O.); akiwase@gunma-u.ac.jp (A.I.); 2Department of Radiation Oncology, Gunma University Graduate School of Medicine, 3-39-22, Showa-machi, Maebashi 371-8511, Gunma, Japan; k-ando@gunma-u.ac.jp (K.A.); tohno@gunma-u.ac.jp (T.O.); 3Gunma University Heavy Ion Medical Center, 3-39-22, Showa-machi, Maebashi 371-8511, Gunma, Japan

**Keywords:** squamous cell carcinoma antigen, cervical cancer, radiotherapy, non-cancer specific elevation, incidence, renal dysfunction

## Abstract

The elevation of the serum squamous cell carcinoma (SCC) antigen unrelated to disease progression occurs during the follow-up of patients with cervical cancer treated with radiotherapy. Although known empirically, the incidence and characteristics of this non-cancer specific elevation in SCC remain unclear. Here, we examined the post-treatment kinetics of SCC in 143 consecutive patients with squamous cell carcinoma of the cervix treated with definitive radiotherapy; in all patients, progression-free disease status was confirmed by periodic monitoring for at least 36 months (median, 61 months). We found that the 5-year cumulative incidence of post-treatment SCC elevation was unexpectedly high at 37.3% (59/143 patients), and that 59.3% (35/59) of event-positive patients experienced multiple events. The median peak SCC level for a given event was 2.0 ng/mL (interquartile range, 1.7–2.9 ng/mL). The multivariate analysis showed that renal dysfunction was associated significantly with a greater incidence of SCC elevation (*p* = 0.046). In addition, the 5-year cumulative incidence of SCC elevation was significantly greater in patients with renal dysfunction than in those without (54.8% vs. 32.9%, respectively; hazard ratio, 2.1 [95% confidence interval, 1.1–4.2]; *p* = 0.028). These data will be useful for monitoring cervical cancer patients treated with radiotherapy.

## 1. Introduction

Cervical cancer causes more than 0.3 million deaths annually worldwide, with a mortality rate that ranks fourth among all cancers [1]. Radiotherapy is the standard definitive treatment for locally advanced cervical cancers [2]. The serum squamous cell carcinoma (SCC) antigen plays a pivotal role in monitoring the local recurrence or metastasis of cervical cancers post-radiotherapy [3]. Evidence suggests that the elevation of serum SCC levels precedes the clinical detection of progressive disease by 2–5 months [4,5,6], and that the intra- or post-treatment normalization of serum SCC level has prognostic significance for a favorable outcome [7,8,9]. SCC, initially named as tumor antigen 4, was first identified in human cervical squamous cell carcinoma tissue by Kato and Torigoe in 1977 [10]. The elevation of serum SCC in cancer patients is due to the passive leakage of the antigen from cancer tissues into the blood [11]. The serum half-life of SCC is 24 h [12]. In accordance with this, the serum SCC level in cancer patients normalizes within 3 days of complete surgical resection of the tumor [13,14].

SCC is expressed not only by cancer cells but also by normal epithelial cells. However, the antigen is not detected in the circulation of healthy adults [11]. During the follow-up of cervical cancer patients after definitive radiotherapy, the elevation of serum SCC unrelated to cancer progression (referred to hereafter as non-cancer specific elevation) is often observed. This non-cancer specific elevation causes severe anxiety for both patients and clinicians and may even lead to excessive examination or unnecessary salvage treatment. However, the incidence and characteristics of this phenomenon remain unclear. Therefore, we investigated the post-treatment kinetics of SCC in patients with cervical cancer treated with definitive radiotherapy, in whom progression-free disease status was confirmed by periodic monitoring.

## 2. Materials and Methods

### 2.1. Study Cohort

This retrospective study enrolled patients who met the following inclusion criteria: (i) newly diagnosed and pathologically confirmed squamous cell carcinoma of the cervix; (ii) staged as IB–IVA based on the International Federation of Gynecology and Obstetrics (FIGO) 2009 staging system; (iii) treated at Gunma University Hospital (Maebashi, Japan) from April 2000 to March 2016; (iv) treated with definitive radiotherapy; (v) no evidence of disease progression confirmed by a bimanual pelvic examination performed every 3–6 months for up to at least 3 years, pelvic magnetic resonance imaging (MRI) performed at 3 months, and computed tomography (CT) from the neck to the pelvis performed at 6 and 12 months, and annually thereafter; and (vi) followed up for at least 36 months.

The details of definitive radiotherapy have been described previously [15]. The first day of radiotherapy is defined as Day 1. The timing of the follow-up CT is modified at the discretion of the attending physician; the examination can be omitted after 5 years without evidence of disease progression, or the examination interval can be shorter than planned in cases under suspicion of disease progression.

### 2.2. Data Collection

The following data were collected from medical records: age, FIGO stage, pre-treatment tumor volume (based on MRI) and nodal status (based on CT and/or ^18^F-fluorodeoxyglucose positron emission tomography), adverse effects (AE) of radiotherapy (based on the Common Terminology Criteria for Adverse Events version 4.0), and serum SCC level.

The serum SCC measurement was performed at 1, 2, 3, 6, 9, and 12 months, and then every 3–6 months thereafter using the chemiluminescent immunoassay-based SCC Abbott Architect kit (F5-Y301-1/J12/R01; Abbott Japan, Tokyo, Japan), with a cutoff of 1.5 ng/mL. The timing of the measurement was modified at the discretion of the attending physician (e.g., for cases with elevated SCC, re-evaluation every 1–2 months was performed until normalization). One event of non-cancer specific SCC elevation was defined either as an increase in SCC above the cutoff value after achieving a post-treatment nadir, followed by a spontaneous decrease below the cutoff, or as an SCC increase that was sustained until the end of follow-up.

Data on the clinical factors that potentially affect non-cancer specific SCC elevation (see the following section for details) were also collected from medical records.

### 2.3. Literature Review on Non-Cancer Specific SCC Elevation

Clinical factors that could potentially affect the occurrence of the non-cancer specific elevation of serum SCC levels were searched using a systematic literature screen. On 7 June 2021, PubMed was searched using the term “squamous cell carcinoma antigen”. For each hit, the title was screened. Next, the abstracts of the title screen-positive papers were screened. The abstract screen-positive papers were then subjected to a full examination, from which clinical factors reported to be associated with the non-cancer specific elevation of serum SCC level were extracted.

### 2.4. Statistical Analysis

A univariate and multivariate analysis was performed using logistic regression in SPSS (version 26; IBM, Armonk, NY, USA). Cumulative incidence was estimated using the Kaplan–Meier method, and the results were compared using the log-rank test in GraphPad Prism8 (GraphPad Software, San Diego, CA, USA). A *p* value of <0.05 was considered statistically significant.

## 3. Results

One-hundred and forty-three patients were eligible for inclusion in the final analysis (Figure 1). The median follow-up period was 61 months (range, 36–126 months). The patient characteristics are summarized in Table 1. The pre-treatment serum SCC level was above the cutoff value in 83.2% (119/143) of patients; this proportion was consistent with a previous study that analyzed the association between serum SCC levels and radiotherapy outcomes for patients with cervical cancer (i.e., 74.6%) [7]. Meanwhile, serum SCC levels normalized at least once within 6 months in 92.3% (132/143) of patients. These results suggest that the dataset is robust.

The post-treatment SCC kinetics for all patients are summarized in Figure 2a. SCC elevation was observed in 59 patients (128 events in total), and the 5-year cumulative incidence was surprisingly high, at 37.3% (Figure 2b). Overall, 59.3% (35/59) of event-positive patients experienced multiple events (Figure 3a). The median peak SCC level for a given event was 2.0 ng/mL (interquartile range, 1.7–2.9 ng/mL) (Figure 3b). The incidence tapered off slightly over time; however, events were observed throughout the follow-up period (Figure 3c). There was no evident seasonality with respect to the time of occurrence (Appendix A). Elevated SCC levels fell to below the cutoff value within 3 months and 6 months in 52.3% (67/128) and 73.4% (94/128) of events, respectively (Figure 3d), whereas the SCC level remained above the cutoff until the end of the follow-up in 12.5% (16/128) of events.

A literature review identified 30 papers reporting clinical factors potentially associated with the non-cancer specific elevation of serum SCC levels (Appendix A, Table 2) [16,17,18,19,20,21,22,23,24,25,26,27,28,29,30,31,32,33,34,35,36,37,38,39,40,41,42,43,44,45]. Based on these data, the association between benign respiratory disease, benign skin disease, or renal dysfunction (RD, defined by an estimated glomerular filtration rate of <60 mL/min/1.73 m^2^ over 3 months [46]) during the follow-up and the occurrence of SCC elevation was analyzed. A univariate analysis showed that RD and benign skin disease were associated significantly with a greater incidence of SCC elevation (*p* = 0.012 and 0.024, respectively) (Table 3). Each skin disease did not show a significant association with the occurrence of SCC elevation (Appendix A). A multivariate analysis showed that RD was associated significantly with a greater incidence of SCC elevation (*p* = 0.046) (Table 3). The 5-year cumulative incidence of SCC elevation was significantly greater for RD-positive patients than for RD-negative patients (54.8% vs. 32.9%, respectively; hazard ratio, 2.1 [95% confidence interval, 1.1–4.2]; *p* = 0.028) (Figure 4a). Non-cancer specific SCC elevation was observed in the first 3–4 years, regardless of the presence or absence of RD (Figure 4a). In this study, 62% (88/143) of patients received platinum-based chemotherapy concomitantly with radiotherapy (Table 1). The use of concomitant chemotherapy did not correlate significantly with the presence of RD (*p* = 0.34) or with the incidence of SCC elevation (*p* = 0.64). Multiple events tended to be more common in RD-positive patients than in RD-negative patients, although the difference was not statistically significant (*p* = 0.095) (Figure 4b). RD had no significant effect on the peak SCC level, the time of the event occurrence, or the duration of events (*p* = 0.49, 0.33, and 0.27, respectively).

SCC levels are presented as the average ± standard deviation or as the median (range), unless stated otherwise.

The AE Grade was based on the Common Terminology Criteria for Adverse Events version 4.0. The odds ratio and *p* values, as assessed by logistic regression, are shown. Clinical factors for which the *p* value in the univariate analysis was <0.3 were entered into a multivariate analysis and examined using a stepwise method.

## 4. Discussion

To the best of our knowledge, this study is the first to report the cancer progression-unrelated kinetics of SCC in cancer survivors. The main findings of this study are as follows: First, the incidence of post-treatment non-cancer specific SCC elevation in patients with cervical cancer treated with definitive radiotherapy was unexpectedly high; approximately one third of the patients experienced the event. Second, RD was an independent factor that increased the risk of non-cancer specific SCC elevation in this population. These data will be useful for monitoring patients with cervical cancer post-radiotherapy.

The patients in this study were followed up for at least 3 years; the median follow-up period exceeded 5 years. Multiple studies have analyzed the radiotherapy outcomes of cervical cancer by employing inclusion criteria compatible with those used for this study; these studies show that the onset of disease progression after 3 years post-radiotherapy is rare [15,47,48]. From this standpoint, the progression-free disease status and the resultant incidence of non-cancer specific SCC elevation identified in this study cohort is highly reliable. If we take the short serum half-life of SCC into consideration [12], the results of the present study indicate that SCC in irradiated tumor tissues is released gradually into the bloodstream as the tissues degenerate, thereby contributing to the disease progression-unrelated elevation of serum SCC observed during post-treatment follow-up. Nevertheless, the onset of this event was not evidently dependent on the tumor burden since neither the tumor volume nor the FIGO stage was associated with the incidence of SCC elevation. Yamamoto et al. investigated brachytherapy-treated prostate cancer tissues and found that anti-tumor immune responses, coupled with the therapeutic effect of radiation on the tumor, are responsible for the disease unrelated, post-treatment elevation of prostate-specific antigen [49]. It is possible that the non-cancer specific SCC elevation observed in radiotherapy-treated cervical cancer patients shares similar biological mechanisms. Another possibility is that there may be a dose–volume correlation, which could explain the event onset. Further research (from both biological and physical perspectives) is needed to investigate this issue more deeply. A recent study indicates that the tumors arising from high-risk human papillomavirus (HPV)-negative cervical dysplasia show favorable outcomes [50]. Another study indicates that preoperative conization is associated with favorable outcomes from a laparoscopic hysterectomy [51]. From these perspectives, the association of serum SCC kinetics with HPV status or that with pathological findings from surgical specimens should be further explored in future studies.

A literature search identified RD as a potential risk factor for non-cancer specific SCC elevation. We confirmed this in a study cohort comprising patients with cervical cancer treated with radiotherapy (hazard ratio, 2.1). Our data also indicate that RD may contribute to multiple event onset. Locally advanced cervical cancer is characterized by the frequent involvement of the parametria, which induces hydronephrosis by causing ureteral stenosis. In addition, platinum-based chemotherapy is a potential risk for RD. These factors may contribute to the greater incidence of non-cancer specific SCC elevation in patients with cervical cancer compared with those with other cancers. Taken together, these data indicate that the post-treatment non-cancer specific SCC elevation observed in cervical cancer patients presenting with RD should be distinguished carefully from cancer progression.

The limitations of this study are as follows. First, the timing of SCC examination was not standardized among participants with non-cancer specific elevation; this might have biased the results. In fact, the management of the cases with persistent serum SCC elevation during post-treatment follow-up was not standardized, and warrants further research. In our department, in general, the patients with elevated SCC were monitored for the antigen levels monthly and subjected to a contrast enhanced CT and/or ^18^F-fluorodeoxyglucose positron emission tomography if the SCC levels were above cutoff for consecutive two tests. Secondly, the association between each skin disease and the occurrence of non-cancer specific SCC elevation was not explored fully, probably due to the small sample size.

In summary, we examined the incidence and characteristics of the non-cancer specific elevation of serum SCC levels in patients with cervical cancer treated with definitive radiotherapy during post-treatment follow-up. The incidence of non-cancer specific SCC elevation in this study cohort was unexpectedly high, and RD was an independent risk factor for event onset. These data will help us to improve the quality of follow-up for cervical cancer patients treated with radiotherapy.

## Figures and Tables

**Figure 1 diagnostics-11-01585-f001:**
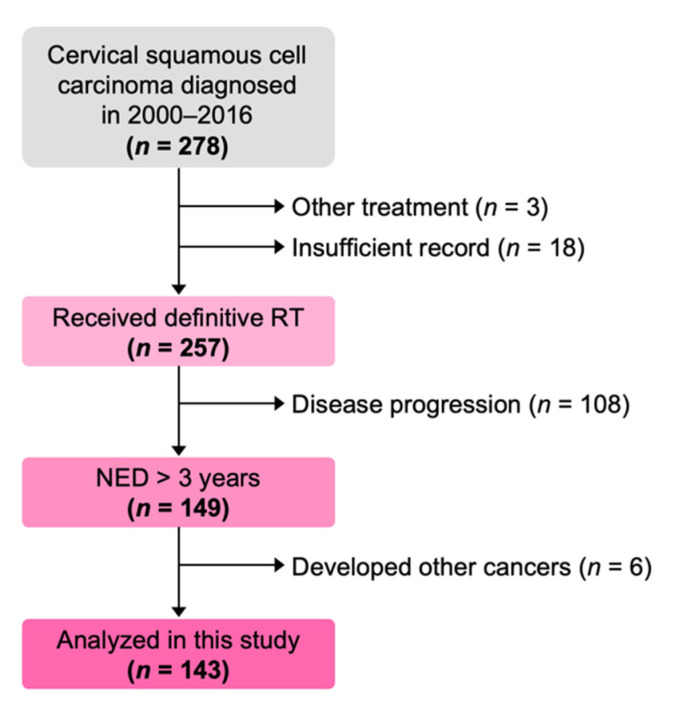
Flow diagram showing patient enrollment. RT, radiotherapy; NED, no evidence of disease progression.

**Figure 2 diagnostics-11-01585-f002:**
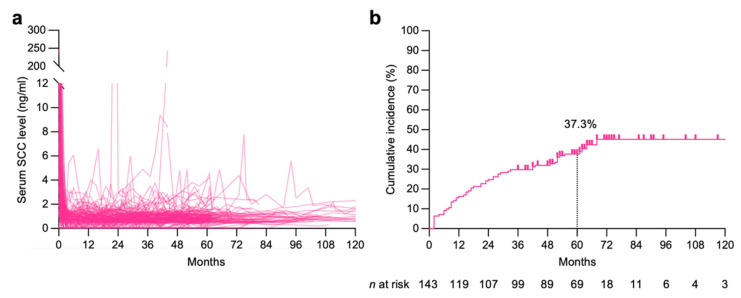
Post-treatment non-cancer specific elevation of serum squamous cell carcinoma (SCC) antigen in patients with cervical cancer treated with definitive radiotherapy (*n* = 143). (**a**) SCC kinetics for all patients. (**b**) Cumulative incidence of non-cancer specific SCC elevation for all patients.

**Figure 3 diagnostics-11-01585-f003:**
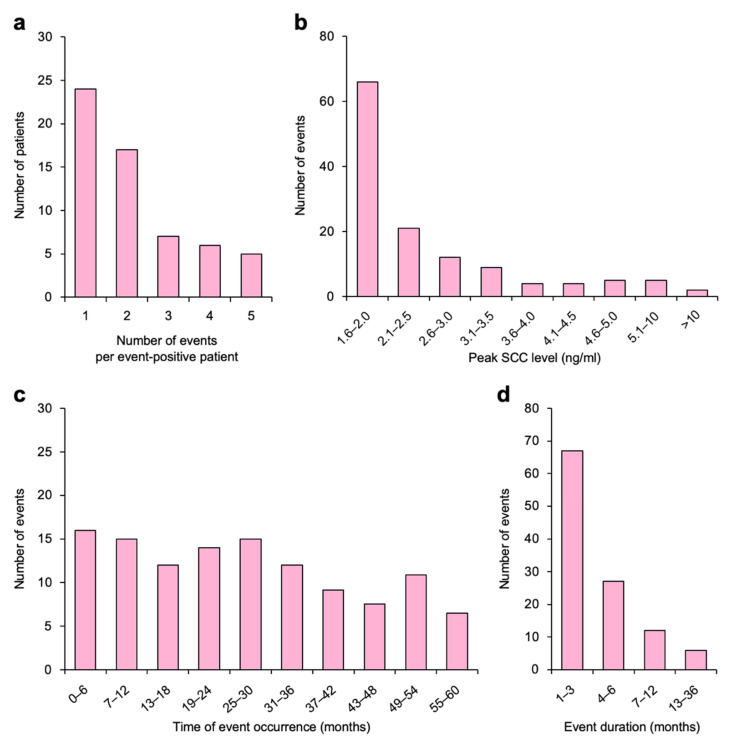
Characteristics of post-treatment non-cancer specific elevation of the serum squamous cell carcinoma (SCC) antigen in patients with cervical cancer treated with definitive radiotherapy. (**a**) Number of events per event-positive patient (*n* = 59). (**b**) Peak SCC level at a given event (*n* = 128). (**c**) Time of event occurrence (*n* = 112). Sixteen events that occurred at >60 months are not shown. (**d**) Duration of a given event (*n* = 112). Sixteen events sustained at the time of final analysis are not shown.

**Figure 4 diagnostics-11-01585-f004:**
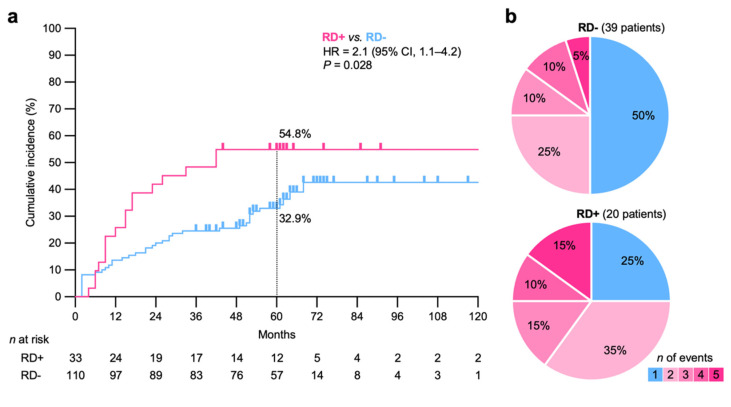
Post-treatment non-cancer specific elevation of serum squamous cell carcinoma (SCC) antigen in cervical cancer patients treated with definitive radiotherapy, stratified according to the presence or absence of renal dysfunction (RD) during the follow-up. (**a**) Cumulative incidence stratified according to RD (*n* = 143). HR, hazard ratio; CI, confidence interval. *p* values were assessed using the log-rank test. (**b**) Number of events per event-positive patient, stratified according to RD (*n* = 59).

**Table 1 diagnostics-11-01585-t001:** Patient characteristics.

Characteristics	*N*
Age	57 (27–88)
FIGO stage	
IB	28 (19%)
II	71 (50%)
III	37 (26%)
IVA	7 (5%)
Tumor volume	
<30 cm^3^	61 (43%)
30–100 cm^3^	57 (40%)
>100 cm^3^	22 (15%)
NA	3 (2%)
Pelvic LN	
Positive	61 (43%)
Negative	82 (57%)
PALN	
Positive	17 (12%)
Negative	126 (88%)
Treatment	
CCRT	88 (62%)
RT alone	55 (38%)

CCRT: concurrent chemoradiotherapy; FIGO: the International Federation of Gynecology and Obstetrics 2009; LN: lymph nodes; PALN: para-aortic lymph nodes; RT: radiotherapy. Age is presented as the median value (range).

**Table 2 diagnostics-11-01585-t002:** Reported clinical factors associated with non-cancer specific elevation of serum SCC.

Clinical Factors	Study Design	*N*	Serum SCC (ng/mL)	Refs.
Respiratory disease				
ARDS	Case report	1	29.5	[16]
COVID-19	Retro MI	252	NA	[17]
Influenza	Case report	1	17.1	[18]
PIE syndrome	Case report	1	8.1	[19]
Various	Retro MI	299	mean, 1.3; range, 0.5–18.3	[20]
Various	Retro MI	89	median, 4.6	[21]
Skin disease				
Atopic dermatitis	Retro MI	30	NA	[22]
Generalized eczema	Retro MI	51	4.1 ± 4.3	[23]
Lichen planus	Retro MI	109	1.3 ± 1.2	[24]
Lichen planus	Case report	1	9.0	[25]
Psoriasis	Retro MI	68	6.5 ± 5.2	[23]
Psoriasis	Retro MI	6	12.7 (1.7–33.3)	[26]
Various	Retro MI	83	NA	[27]
Various	Retro MI	72	range, 3.5–117.7	[28]
Various	Retro MI	66	range, 1.4–73.2	[29]
Renal dysfunction				
Diabetic nephropathy	Retro MI	77	1.2 (0.5–1.8)	[30]
Various, with dialysis	Retro MI	94	NA	[31]
Various, without dialysis	Retro MI	539	median, 0.8	[32]
Pregnancy-related events				
Amniotic embolism	Retro MI	4	112.0 ± 169.4	[33]
Follicle stimulation	Retro MI	42	2.0 (1.1–17.8)	[34]
Normal pregnancy	Retro MI	56	1.7 (0.8–17.0)	[35]
Normal pregnancy	Prospective	12	range, 0.1–4.3	[36]
Pediatric diseases				
Asthma	Retro MI	32	2.1 (1.4–3.3)	[37]
Atopic dermatitis	Prospective	96	NA	[38]
Atopic dermatitis	Retro MI	95	NA	[39]
Others				
Arterial puncture	Retro MI	13	NA	[40]
Chromium hexavalent	Retro MI	115	1.2 ± 1.2	[41]
Infectious spondylitis	Case report	1	2.2	[42]
Inverted papilloma	Prospective	30	3.9 (IQR, 2.3–7.9)	[43]
PAH	Retro MI	136	range, 0.2–2.4	[44]
Teratoma	Case report	1	6.4	[45]

ARDS: acute respiratory distress syndrome; COVID-19: coronavirus disease 2019; IQR: interquartile range; MI: mono-institutional; NA: not assessed; PAH: polycyclic aromatic hydrocarbons; PIE: pulmonary infiltration with eosinophilia; Refs: references; Retro: retrospective; SCC: squamous cell carcinoma antigen.

**Table 3 diagnostics-11-01585-t003:** Association between clinical factors and the occurrence of non-cancer specific SCC elevation.

Clinical Factor		SCC Elevation		Univariate		Multivariate	
		+	−	Odds Ratio (95% CI)	*p*	Odds Ratio (95% CI)	*p*
Age						NA	
	<39	8	7	Ref			
	40–49	12	14	0.75 (0.21–2.68)	0.65		
	50–59	14	25	0.49 (0.14–1.63)	0.24		
	60–69	15	22	0.59 (0.17–1.99)	0.40		
	≥70	10	16	0.54 (0.15–1.97)	0.35		
FIGO stage						NA	
	IB	12	16	Ref			
	II	29	42	0.92 (0.38–2.23)	0.85		
	III	14	23	0.81 (0.29–2.20)	0.68		
	IVA	4	3	1.77 (0.33–9.47)	0.50		
Tumor volume							
	<34.3 cm^3^	25	46	Ref			
	≥34.3 cm^3^	32	37	1.70 (0.86–3.37)	0.12	1.38 (0.65–2.93)	0.39
Pelvic LN						NA	
	Negative	34	48	Ref			
	Positive	25	36	0.98 (0.50–1.92)	0.95		
PALN						NA	
	Negative	52	74	Ref			
	Positive	7	10	0.99 (0.35–2.78)	0.99		
Treatment						NA	
	CCRT	24	31	Ref			
	RT	35	53	0.85 (0.43–1.68)	0.64		
AE, skin							
	Grade 0	51	64	Ref			
	Grade ≥1	8	20	0.50 (0.20–1.23)	0.13	0.48 (0.18–1.29)	0.14
AE, GI						NA	
	Grade 0–1	37	47	Ref			
	Grade ≥2	22	37	0.75 (0.38–1.49)	0.41		
AE, GU						NA	
	Grade 0	38	58	Ref			
	Grade ≥1	21	26	1.23 (0.60–2.49)	0.56		
AE, hematology							
	Grade 0–1	17	35	Ref			
	Grade ≥2	42	49	1.76 (0.86–3.59)	0.11	1.53 (0.69–3.38)	0.28
Respiratory disease						NA	
	Negative	49	72	Ref			
	Positive	10	12	1.22 (0.49–3.05)	0.66		
Skin disease							
	Negative	47	78	Ref			
	Positive	12	6	3.31 (1.16–9.43)	**0.024**	2.63 (0.88–7.84)	0.083
Renal dysfunction							
	Negative	39	71	Ref			
	Positive	20	13	2.80 (1.25–6.23)	**0.012**	2.38 (1.01–5.61)	**0.046**

AE: adverse effects; CCRT: concurrent chemoradiotherapy; CI: confidence interval; FIGO: International Federation of Gynecology and Obstetrics 2009; GI: gastrointestinal; GU: genitourinary; LN: lymph nodes; NA: not assessed; PALN: para-aortic lymph nodes; Ref: reference; RT: radiotherapy; SCC: squamous cell carcinoma antigen.

## Data Availability

The data are not publicly available due to the study protocol approved by the Institutional Ethical Review Committee of Gunma University Hospital.

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
