# Peer review of "The Non-Cancer Specific Elevation of the Serum Squamous Cell Carcinoma Antigen during the Post-Radiotherapy Follow-Up of Cervical Cancer Patients"

_diagnostics, 2021, doi:10.3390/diagnostics11091585_

Round 1
Reviewer 1 Report
I read with great interest the manuscript, which falls within the aim of this Journal. In my honest opinion, the topic is interesting enough to attract the readers’ attention. Nevertheless, authors should clarify some points and improve the discussion, as suggested below.
Authors should consider the following recommendations:
- Manuscript should be further revised in order to correct some typos and improve style.
- I would recommend to stress novel pieces of evidence about high-risk HPV-negative high-grade cervical dysplasia, which seems to have more favorable outcomes than patients with documented high-risk-HPV infection (PMID: 33514481) as well the potential role of pre-operative conization before radical hysterectomy for early-stage cervical cancer (PMID: 32320800).
Author Response
I read with great interest the manuscript, which falls within the aim of this Journal. In my honest opinion, the topic is interesting enough to attract the readers’ attention. Nevertheless, authors should clarify some points and improve the discussion, as suggested below.
Response:
We sincerely thank the reviewer for evaluating our manuscript and for the encouraging comments. According to the suggestion, we revised the manuscript as follows.
Authors should consider the following recommendations.
Response:
Thank you for the comment. The manuscript was revised as follows.
Manuscript should be further revised in order to correct some typos and improve style.
Response:
We thank the reviewer for the valuable suggestion. According to the suggestion, the manuscript was subjected to a professional English editing by Bioedit Ltd. (UK). The editing certificate is attached in the Word file.
I would recommend to stress novel pieces of evidence about high-risk HPV-negative high-grade cervical dysplasia, which seems to have more favorable outcomes than patients with documented high-risk-HPV infection (PMID: 33514481) as well the potential role of pre-operative conization before radical hysterectomy for early-stage cervical cancer (PMID: 32320800).
Response:
We thank the reviewer for the valuable suggestion. According to the suggestion, the necessity of future exploration of serum SCC kinetics was discussed in relation to the novel evidence from PMID:33514481 and PMID:32320800. The following sentences was added in Discussion (lines 210–215). A recent study indicates that the tumors arising from high-risk human papillomavirus (HPV)-negative cervical dysplasia show favorable outcomes (PMID:33514481). Another study indicates that preoperative conization is associated with favorable outcomes of laparoscopic hysterectomy (PMID:32320800). From these perspectives, association of serum SCC kinetics with HPV status or that with pathological findings form surgical specimens should be further explored in future studies. PMID:33514481 and PMID:32320800 were added as Reference # 50 and #51, respectively.

Reviewer 2 Report
This study evaluated non-cancer specific elevation of serum SCC antigen after definite radiotherapy for the treatment of cervical cancer. Renal dysfunction was significantly associated with non-cancer specific elevation of serum SCC antigen.
Major comments
1. As your literature review, many studies have been published regarding non-cancer specific elevation of serum SCC. What was the new findings compared to previous study?
2. How did you define the renal dysfunction?
3. Detail information of skin diseases should be described and evaluated the association between each skin diseases and elevation of serum SCC antigen.
4. If non-cancer specific elevation of serum SCC antigen was persisted, how did you manage the patients or how often did you conduct imaging studies?
Author Response
This study evaluated non-cancer specific elevation of serum SCC antigen after definite radiotherapy for the treatment of cervical cancer. Renal dysfunction was significantly associated with non-cancer specific elevation of serum SCC antigen.
Response:
We sincerely thank the reviewer for evaluating our manuscript. According to the suggestion, we revised the manuscript as follows.
Major comments
- As your literature review, many studies have been published regarding non-cancer specific elevation of serum SCC. What was the new findings compared to previous study?
Response:
We thank the reviewer for the critical comments. The new findings of this study are that this study demonstrated for the first time the cancer progression-unrelated kinetics of SCC in cancer survivors. This is clearly in contrast to the studies listed in Table 2 (i.e., the summary of our literature review) because those studies analyzed benign diseases in non-cancer patients. The novelty of this study was clarified in lines 183–184.
- How did you define the renal dysfunction?
Response:
We thank the reviewer for the important comment. In this study, the renal dysfunction was defined as an estimated glomerular filtration rate of <60 ml/min/1.73 m2 over 3 months based on KDIGO 2012 Clinical practice guideline (Kidney Int. Suppl. 2013, 3, 1–150). This was clarified in lines (141–142) and in Reference #46.
- Detail information of skin diseases should be described and evaluated the association between each skin diseases and elevation of serum SCC antigen.
Response:
We thank the reviewer for the valuable suggestion. According to the suggestion, we showed the detail information of skin disease and performed additional analyses to investigate the association between each skin disease and non-cancer specific elevation of serum SCC. Each skin disease did not show any significant association with the occurrence of non-cancer specific elevation of serum SCC. This was probably due to the small number of cases, and therefore, noted as the limitation of this study. These data are added in lines (145–146, 233–235) and as Supplementary Table S1.
- If non-cancer specific elevation of serum SCC antigen was persisted, how did you manage the patients or how often did you conduct imaging studies?
Response:
We thank the reviewer for the valuable suggestion. The management for the cases with persistent serum SCC elevation during post-treatment follow-up is not standardized, warranting further research. In our department, in general, the patients with elevated SCC are monitored for the antigen levels monthly, and subjected to contrast enhanced CT and/or 18F-fluorodeoxyglucose positron emission tomography if the SCC levels were above cutoff for consecutive two tests. This was added in lines 229–233.
